# Reply to Barron et al. Comment on “Cook, B.; Mulgan, N. Targeted Mop up and Robust Response Tools Can Achieve and Maintain Possum Freedom on the Mainland. *Animals* 2022, *12*, 921”

**DOI:** 10.3390/ani13213343

**Published:** 2023-10-27

**Authors:** Briar Cook, Nick Mulgan

**Affiliations:** 1Tasman District Council, Richmond 7020, New Zealand; briar.cook@tasman.govt.nz; 2Zero Invasive Predators, Wellington 6012, New Zealand

## 1. Introduction

We are pleased that our paper [1] on possum elimination has caught the attention of researchers from Landcare Research Manaaki Whenua [2]. The progress of science requires critical thinking and collaboration, and we would be interested in collaborating on further research.

The paper reports on a project to remove mammalian predators from a mainland landscape and defend against reinvasion, focussing on possums (*Trichosurus vulpecula*) in the two years following the initial control.

Regrettably, there was an error in the paper. We correct this below. Some aspects of the admittedly complicated situation were not explained well. We attempt to elucidate these also. As requested, we have included a modelled estimate of possum numbers.

Our overall conclusion about the success of possum elimination remains the same.

## 2. Response

### 2.1. Spatial Extent

There appears to be some confusion, which we have no doubt contributed to, about the terrain in the project area (see Figure 1).

Overall, 11,642 ha is the approximate total catchment area, restricted to the east and south of the Perth and Barlow rivers, and otherwise bounded by the Southern Alps/Kā Tiritiri-o-te-Moana.

Our “8700 ha suitable possum habitat” in the abstract is incorrect. We regret this. The value of 8700 ha was the total area treated in the first toxin drop, which included areas across the rivers to minimise immediate incursion. The correct figures are 3191 ha of vegetation and cover [3], so good possum habitat [4], and a further 2572 of tussock and alpine grasses without cover, called “marginal” habitat below, to make a total of 5763 ha possum habitat. The remaining area, 5879 ha, is bare rock, snow, and ice.

### 2.2. Area Monitored

The 142 lured trail cameras covered the good habitat and some of the marginal habitat, approximately evenly distributed at 1 per 500 m, on lines 700 m apart, except for one area of approximately 130 ha that was not safely accessible on foot, and one tributary valley that is above the tree line. The exact placement of the cameras was dictated by access and operational efficiency, so a regular grid was not possible. The goal was detection, not calibration to an index, so the lack of an exact grid was not considered critical. Following control, possum home ranges probably did expand [5], meaning that more than one camera overlapped a possum’s home range. No claim is made that the cameras are sufficiently spaced that they would be measuring possums independently.

Overall, 87% of the good habitat and 18% of the marginal habitat were within 430 m of the 142 cameras (430 m is the half-diagonal of a 500 m by 700 m grid).

**Figure 1 animals-13-03343-f001:**
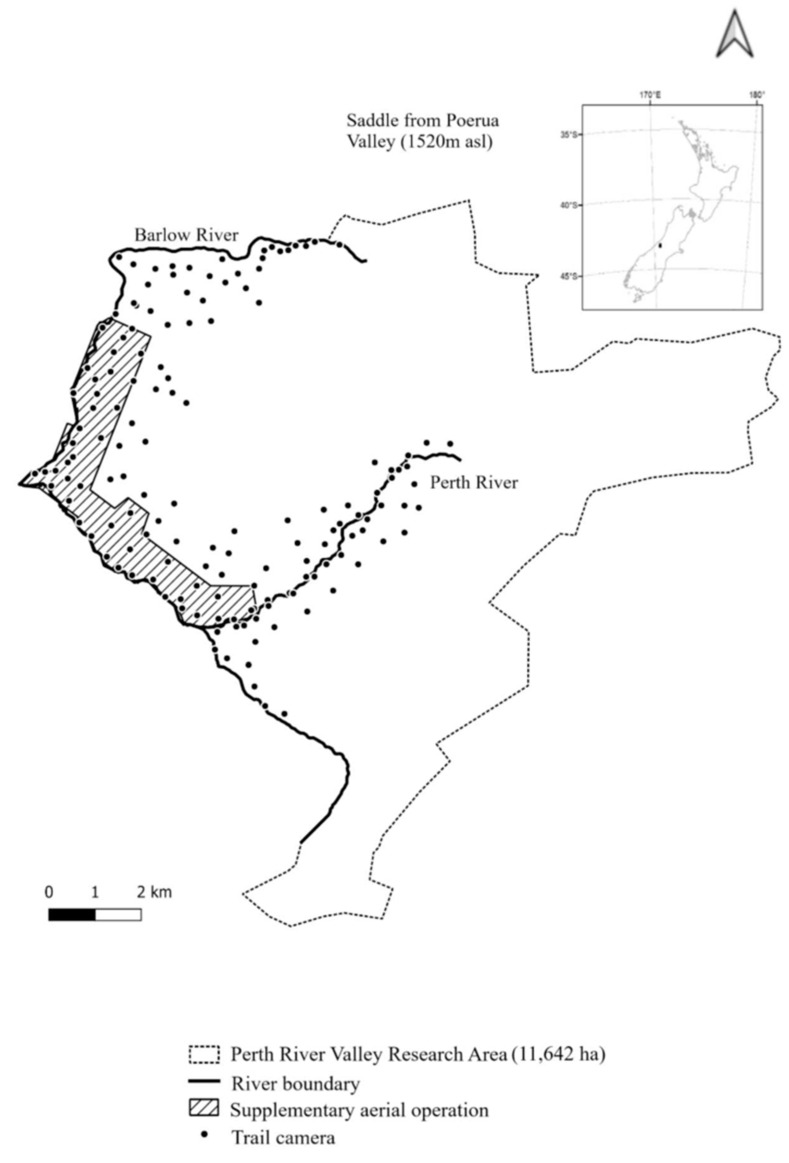
Study site showing trail camera locations, protective river boundaries, and the supplementary aerial treatment coverage.

We hypothesised that possums will only occupy the marginal habitat if there is some neighbouring good habitat within their activity area, or if they are pushed into it by high intraspecific pressure. Possums in low-density environments typically include some high-quality denning areas [5,6].

Other detection and catching efforts, including additional cameras, searches with a possum dog, and cages, were mentioned in the paper but not detailed. Because this was a novel type of project, detection tools were being modified and adapted to increase efficacy in real time. Approximately 270 further-lured cameras were deployed between July 2019 and July 2020 on an ad hoc basis in areas where rats were detected. Almost all were in the good possum habitat. Approximately 12 cameras were similarly deployed where possums were detected, in the Barlow River Headwaters and Upper Perth Valley (see Figure 2), again mostly in good habitat. Dog searches totalled over 350 km. From mid-2021, the extra cameras were progressively replaced with overhead infra-red AI cameras then under development.

Barron et al. [2] are correct that half the area was not monitored. Half the area is rock and ice. Possums may travel over some of this terrain, but, with no food available, the fraction of time spent there will likely be small [7], and no individuals will be missed by not targeting it.

### 2.3. Detections

The progress of detections across the 142 permanent cameras, with all captures shown in Table 1.

In addition to the captures, any remaining possums may have been killed in the supplementary toxin operations that targeted ship rats.

Sporadic detections in good habitat in the Mid-Perth Valley from 2021 to 2023 and Upper Perth Valley from 2022 to 2023 showed that at least one possum had survived or come across the borders and travelled as far as this. Detections continued in the Barlow Headwaters, supporting the hypothesis in the paper that possums were entering from the Poerua Valley to the north (Figure 1).

### 2.4. Detectability Calculation

Barron et al. [2] allege that the analysis we included to measure detectability is flawed because we have not determined the remaining number of animals. The simplest explanation for the Upper Perth Valley timeline in Figure 3 in [1], with detections before the two captures and none following for over a year, is that the two possums captured, and no more, were responsible for the detections. This is the context in which ‘exactly’ was used. An alternative explanation that more than two animals were detected and that, following the capture of the first two, the remainder were stochastically not detected, or changed their behaviour so as not to be detected, is possible, but unlikely (the additional cameras also had no detections over this time period.

Barron et al. [2] state that this detectability calculation was not linked to elimination. That is correct. No such claim was made.

### 2.5. Detectability Comparison with the Literature

We used *g*_0_ = 0.17 per night for a camera lured with mayonnaise, dispensed daily. This was not well-explained. There was no published value for such a camera/lure combination. We took the value from Ball et al. [8] and adapted it. Ball estimated *g*_0_ = 0.12 with a range of 0.06 to 0.24 for the encounter rate of a possum at a trap. With the extra reward of food daily, we estimated that the value might increase by 0.05 to 0.17. This value is, of course, speculative. An estimate of the uncertainty is from Ball’s minimum to above their maximum, perhaps 0.06 to 0.30.

Ball et al. have σ = 63 m, or a 95% home range of 7.5 ha. *g*_0_ values in the model below are scaled with the inverse of the home range.

### 2.6. Model

The difficulty with modelling the project is in applying the existing models, such as those suggested by the commenters, to possums in the limit of low abundance. Nevertheless, we input the detection count into SECR [9] function *ENRM* [10] and solve for the number of animals. The errors in the results are estimated by Monte Carlo draws from PERT distributions of the input parameters [11]. This method of error estimation can average out uncertainties, so the uncertainty ranges may be biased narrow.

### 2.7. Assumptions

Whyte et al. [12] show that when density is high, home ranges can be 1 to 2 ha. When density is less than or of the order of 1 per ha, home ranges increase to 5 to 10 ha or more. Evidence in the paper, and elsewhere [13], shows that a lone possum may move over a larger area of at least 40 ha in a few months, so that the concept of home range may not even apply. In the paper, a pair of possums may be occupying an area of the order of only 6 ha for 2 months.

Overall, Ball’s value of 7.5 ha seems a reasonable median, but with a considerable range. We estimate 3 to 50 ha.

We assume the marginal habitat counts as 50% of good habitat, so the effective good habitat is 4477 ha.

## 3. Results

The estimated numbers of possums are shown in Table 2.

The net result is that the population was being held at a level of approximately 0.003 per ha averaged over the effective good habitat (4477 ha) and zero at least temporarily over areas on the scale of 500 to 1000 ha.

## 4. Conclusions

We are still confident that elimination in the Perth Valley was achieved and maintained.

Again, we are grateful for the commenters’ interest. The advancement of science requires critical thinking and collaboration, and we would be interested in collaborating on further research.

## Figures and Tables

**Figure 2 animals-13-03343-f002:**
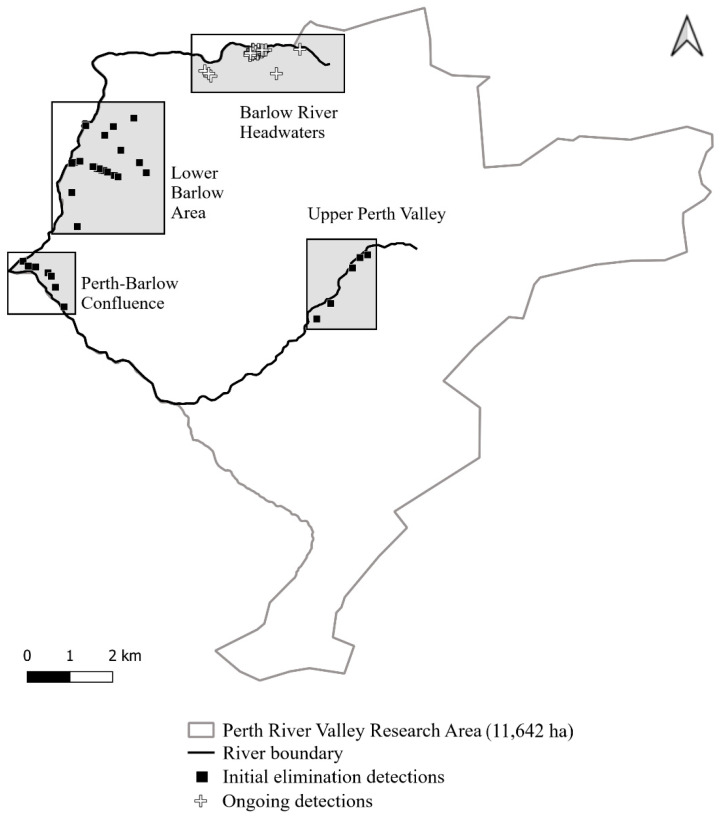
Possum detections in the Perth Valley from 25 August 2019 (first detection following site wide aerial 1080) to 31 August 2020. Shaded squares indicate the discrete areas of the site discussed in the text.

**Table 1 animals-13-03343-t001:** Detections on Permanent Cameras and All Captures.

	2019	2020	2021
Distinct camera-days detecting	26	31	21
Distinct Cameras detecting	13	13	8
Captures	0	7	2

**Table 2 animals-13-03343-t002:** Population Estimates from Model.

	2019	2020	2021
Estimated possums	15	12	10
2.5 percentile	10	8	7
97.5 percentile	28	22	18

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
