# Peer review of "Reply to Barron et al. Comment on “Cook, B.; Mulgan, N. Targeted Mop up and Robust Response Tools Can Achieve and Maintain Possum Freedom on the Mainland. Animals 2022, 12, 921”"

_animals, 2023, doi:10.3390/ani13213343_

Round 1

Reviewer 1 Report

Comments and Suggestions for Authors

To my knowledge, this is one of the first situations, where a paper in Animals got a comment and the authors are answering the comment. As this is normal practice in science, such a “Reply” must be welcome, especially when it reveals some mistakes in the formerly published paper. Possibly, the Title requires modification.

Therefore, my comments on the “Reply” are short; some changes are required:

1.       “Comment” itself must be cited – I prefer to see it in Line 11, as [1] Barron, M.; Anderson, D.P.; Norbury, G.; Warburton, B. Comment on Cook, B.; Mulgan, N. Targeted Mop up and Robust Response Tools Can Achieve and Maintain Possum Freedom on the Mainland. Animals 2022, 12, 921. Animals 2023, 13, 1840. https://doi.org/10.3390/ani13111840

2.       Reference [1] number must be also presented at Line 60 – the reader now has no idea, what is “Barron et al.”

3.       For the readers not to look at the first publication, a map on the study site would be helpful, even if this could be just a combination of the previously published maps

I feel Reply is written correctly. However, none of the reviewers from the side will be able to give 100% evaluation, which ones – authors or commenters – are correct. Therefore, I recommend to publish “Reply” in animals.

I am not sure if Animals need to post a correction of the formerly published article.

Author Response

1.       comment cited as [2]

2.       [2] referenced throughout

3.       Previous Fig 1 included

Reviewer 2 Report

Comments and Suggestions for Authors

The authors have sufficiently replied to the Barron et al. (2023) comment on their article. Some further clarifications are needed though.

Major comment

As the authors include corrections and additions to their article, they should issue a correction attached to the article for consistency, if possible, after agreement with the editorial office.

Specific comments

Line 10: Cite your article here. And include it to the references list.

Line 33: Do you mean the residual area of the total area? Please clarify.

Line 60: Give the reference here, say Barron et al. [5], and include it in the references list. Amend throughout.

Lines 60-61: Is it impossible for possums to be found on rock and ice? Please give some more information with references.

Line 73: Figure 1 in the paper? Please clarify.

Line 77: Figure 3 in the paper? Please clarify.

Author Response

Major comment: I think that the reply to comment is sufficient as correction, but will correct if editor insists.

Line 10: article added as [1]

Line 33: Yes - clarified

Line 60: added as [2], amended

Lines 60-61: Unlikely to be major component of time - reference added

Line 73: clarified

Line 77: clarified